# How peer mechanism impacts loan repayment in a Self-help group?: An empirical study in India

**Nishi Malhotra**  *

Assistant Professor, IIM Sambalpur, Odisha, India

* nishim@iimsambalpur.ac.in

## Abstract

Women's empowerment through financial inclusion is an important sustainable development goal. Formal institutions and banks are hesitant to lend to the poor due to the lack of collateral and information asymmetry. The self-help groups address this gap through a social contract based on social capital and joint liability. The joint liability gives rise to a peer mechanism in the form of peer selection, monitoring and peer sanctions. This study examines how peer group in West Bengal, India, using data from 400 members and ordered logistic regression. As per the findings, peer selection reduces adverse selection, and peer monitoring mitigates moral hazard among the members of the group. Further leadership and access to technology strengthen the repayment behaviour. However, excessive peer sanctions can lead to increased misuse of funds. The study aims to highlight the potential limitations of the peer mechanism in reducing information asymmetry and increasing repayment of loans with policy implications.

## 1. Introduction

> *Credit is the last hope left to people at the bottom.*
>
> *When everything fails, credit becomes the last straw."—*
>
> *Muhammad Yunus, Banker to the Poor:*
>
> *Micro-Lending and the Battle Against World Poverty.*

Globally, approximately 1.3 billion people are financially excluded due to a lack of physical collateral and information asymmetry regarding the creditworthiness of borrowers [1]. Commercial banks and financial institutions lack adequate information about the creditworthiness of low-income households. Hence, these banks and formal financial institutions are cautious about lending to individuals at the bottom of the pyramid. Before the 1990s, most lending initiatives operated as charity and donation-based. In the post-liberalisation era, many of these initiatives transformed

**Data availability statement:** All relevant data are within the paper and its Supporting Information files. Data set is also accessible via figshare: https://doi.org/10.6084/m9.figshare.30912965.

**Funding:** The author(s) received no specific funding for this work.

**Competing interests:** I have no conflicting interest.

into sustainable finance intermediaries that could accept deposits and credit [2]. Early community lending initiatives, such as Rotating Savings and Credit Associations (ROSCAs), which emerged in the 1720s, were designed to alleviate poverty by pooling savings and rotating loans among members. Historical evidence shows that similar models emerged in China (Kongsi), Malaysia (Hui), and South Korea, Ke, Yokai [3] These initiatives relied on cohesion, solidarity, and interpersonal trust but faced various limitations, including information asymmetry and credit risk, [4]. Moreover, these ROSCAs were informal arrangements without legal recourse, making it extremely difficult to recover credit if a member defaults, [5]. These community lending initiatives were susceptible to mismanagement and fraud without formal supervision and oversight.

In the 19th century, the Raiffeisen German Credit Union Cooperative model enabled farmers to pool resources under unlimited group liability. These groups were also formed from a community of a few tightly knit community members. However, as modernisation and newer banking models emerged, the Raiffeisen German Credit Union Cooperative evolved into modern banks with limited liability. Additionally, despite peer selection, these groups were plagued by adverse selection and moral hazard. Thus, though this financial inclusion model was suited for a pre-agrarian society with information asymmetry, it failed in the modern-day, technically advanced banking [6]. By the end of the 19th century, information asymmetry was a significant challenge in financial inclusion, [7]

In response, yet another milestone was the emergence of Mohd Yunus' Grameen Bank model in Bangladesh. It aimed to provide credit to people at the bottom of the pyramid based on peer trust and social capital. Grameen banks promoted regular savings among the women savers [8]. This community initiative aimed to ensure financial inclusion for the people at the bottom of the pyramid through social capital. In a developing economy like Bangladesh, the Grameen Bank financial inclusion model emerged as a landmark in combating information asymmetry through peer selection and monitoring. Later on, based on its success in terms of repayment rates, this initiative was replicated as self-help groups in India, [9]. Another replica of the Grameen Bank model was village-level banking and community-based microfinance, supported by NGOs, which emerged in the 1980s. This community lending initiative was again fraught with elite domination and limited resources [10]. In the Indian scenario, till the 1990s, banking focused on social emancipation through priority sector lending, state subsidy and donation. After the banking reforms, rural banking needed to be more sustainable [11]. Individual lending by microfinance institutions emerged as another business model, whereby the microfinance institutions provide loans at extremely high interest rates to people experiencing poverty at the bottom of the pyramid. This model led to the vicious circle of poverty for the poor and higher non-performing assets for the financial institutions [12]

In 1998, a self-help group bank linkage scheme was introduced in India to facilitate financial inclusion through social capital [13,14],. A self-help group refers to a group of homogeneous people who come together to save regularly and provide loans to other members in the group. Based on their credit history and performance, the

self-help group bank linkage programme connects self-help groups with banks. linkage. The Self-Help Group Bank Linkage (Deendayal Antodaya Yojana, National Rural Livelihood Mission) has consistently reported high repayment rates. This programme is based on complex interactions among banks, self-help groups, and non-government organisations. This initiative has helped to achieve financial inclusion by leveraging a peer mechanism. Within the ambit of this programme, members are peer-selected. This initiative is embedded in social capital through professional and community networks that foster joint liability [15]. Through assortative matching, peer selection facilitates borrower screening, thereby reducing adverse selection. Because the social contract is based on joint liability, these groups work through peer mechanisms to mitigate information asymmetry [16]. Joint liability in a group initiative fosters peer monitoring. As part of group liability, defaulting members are subject to peer sanctions, which helps promote financial sustainability within groups. Various studies highlight that self-help groups in India have repayment rates of more than 95% and have been instrumental in reducing poverty in India [17–19]. This success in achieving financial inclusion through a self-help group is attributed to the peer mechanism and social capital.

Despite the theoretical relevance, the empirical evidence on the impact of the peer mechanism in mitigating adverse selection and moral hazard in a self-help group financing initiative is limited. The present study aims to investigate the role of peer mechanisms in reducing adverse selection and moral hazard.

## 2. Literature review

Globally, approximately 1.3 billion people are financially excluded from the formal financial system and live in abject poverty, [20] The marginalised sections of society often lack physical collateral, and banks are hesitant to lend to individuals with low incomes due to a lack of information about their creditworthiness, particularly those experiencing poverty [21] (Dhungana et al Due to higher monitoring and screening costs for low-income individuals who lack a credit history, banks are reluctant to provide them with loans [21]. The literature refers to the problem as mission drift, whereby the banks prioritise lending to the wealthier sections of society, while neglecting the poor [22] Microfinance has played an important role in the financial inclusion and empowerment of poor people [23,24]. In the post-COVID-19 environment, social finance has emerged as a panacea to combat financial exclusion [25]. From a philosophical perspective, social finance is key for social development, though it conflicts with the objective of financial sustainability and wealth maximisation for the banks [26–28]

### Evolution of microfinance for the financial inclusion of the poor

Until the 1990s, state intervention in donation and subsidy-based finance played a key role in financially including people experiencing poverty, [29]. This poverty lending approach involved government-funded credit at subsidised rates, [30]. Since the 1990s, there has been a paradigm shift toward sustainable finance, aiming to ensure the inclusion of low-income individuals without compromising the sustainability of financial institutions. A moot question till the 1990s was whether access to finance for people with low incomes is affordable. The concept of community lending formats based on social capital gradually emerged as a substitute for physical collateral, enabling people with low incomes to access finance [31]. Due to access to local market information, the microfinance institutions and self-help groups are more effective in achieving financial inclusion, [32].

From a sociological perspective, the main factors impacting the financial behaviour of poor people are social norms and culture [33]. During this period, several microfinance models leveraging social capital emerged. One of the most notable microfinance formats based on social capital was ROSCAs, which involved pooling the savings resources and rotating them as borrowings among the group members. This initiative is based on trust and operates through social pressure. It fosters social cohesion and mutual support among group members. ROSCAs (Rotating Savings and Credit Associations) were informal savings institutions that did not offer any interest to the savers and were vulnerable to the risk of strategic default. Due to the risk of moral hazard cropping from information asymmetry, the growth of ROSCAs declined,

[34]. Similarly, cooperative societies provided credit and financial services to the members at a lower rate. These societies are democratically controlled institutions designed to meet the needs of their members. These cooperative societies suffered from moral hazards, the risk of strategic default by the members, and weaker controls. These societies failed due to governance issues and political interference, resulting in higher non-performing assets and increased fraud [35]. One of the milestones in microfinance was the setting up the Grameen Bank, which Muhammad Yunus founded in Bangladesh, aimed at providing microfinance to the poor women at the bottom of the pyramid. As per Grameen Bank, credit is extended based on social capital, [8]. Various village-level institutions, banks, and institutions also emerged in rural areas, providing loans from the pooled savings, [36]. Despite several initiatives due to the information asymmetry regarding the creditworthiness of people with low incomes, these models had limited success, with Grameen Bank a notable exception. Self-help groups based on social capital emerged as vehicles for the social emancipation of low-income people, [37]. It enabled people with low incomes without a credit history to access microfinance, [38,39]. SEWA by Ela Bhatt was the first self-help group, enabling the poor and marginalised to access finance, [15,40]. India adopted a graduation approach to financial inclusion through the self-help group bank linkage programme. Beginning with NGOs like MYRADA in 1980, which promoted self-help groups, the approach was institutionalised with the NABARD self-help group linkage programme in 1982, [13]. This initiative allowed the poor and marginalised to generate social capital through networked relationships, bonding, and bank linkages. Moreover, policy programmes such as Swarnjayanti Gram Yojana further reinforced the provision of international recognition at the 1997 World Microcredit Summit [41]. Furthermore, state-promoted initiatives such as Kudumshree Kerala and Velegu in Andhra Pradesh have demonstrated how small savings and credit groups can evolve into platforms for livelihood generation and empowerment. The subsequent graduation of self-help groups into policy initiatives, such as Jeevika in Bihar and Tripti in Odisha, led to scaling up these community initiatives into federations and clusters. Eventually, these initiatives culminated in Deen Dayal Antodaya Yojana under the National Rural Livelihood Mission. These milestones illustrate the evolution of small informal savings groups into bank-linked intermediaries [42]. These self-help groups demonstrated high repayment rates and efficient loan disbursal. They emerged as a panacea to the problem of financial exclusion of the poor and marginalised sections of society due to information asymmetry.

Table 1 provides details regarding the evolution of microfinance. Information asymmetry increases the likelihood of adverse selection, whereby the banks might select members with a higher probability of default, [43]. Theoretical models of information asymmetry have emphasized that banks might misinterpret the risk profile of borrowers. This information

**Table 1. Evolution of microfinance (Source: Author's own work).**

| Period | Milestone/Model | Key features | Impact on financial inclusion |
|---|---|---|---|
| Pre-1900s | Informal ROSCAs (e.g., Hui, Kongsi, Ke) | Rotating savings among trusted community members | Provided basic access to credit; limited scale and sustainability. |
| Mid-1800s | Raiffeisen Credit Cooperatives (Germany) | Mutual aid, unlimited liability, rural focus | Enabled farmers to pool resources; faced governance and scalability issues. |
| 1970s–1980s | Grameen Bank Model (Bangladesh) | Group lending | Revolutionised microcredit; high repayment rates among poor women |
| 1980s–1990s | NGO-led SHGs (e.g., MYRADA, SEWA) | Community-based savings and credit groups | Empowered women, built social capital, informal but impactful |
| 1992 | NABARD SHG-Bank Linkage Programme (India) | Formal linkage of SHGs with banks | Institutionalised microfinance; improved access to formal credit |
| 2000s | Microfinance Institutions (MFIs) | Individual lending, commercial orientation | Scaled outreach; concerns over high interest rates and over-indebtedness |
| 2010s–2020s | Digital Microfinance & Fintech | Mobile banking, digital KYC, and credit scoring | Enhanced reach and efficiency; reduced transaction costs and barriers |
| Present | Integrated Livelihood & Financial Models | SHGs linked to skill development, insurance, pensions | Holistic empowerment; financial inclusion as part of broader development goals |

asymmetry underlies credit rationing among rural borrowers. Within self-help groups, members select each other and are, by definition, already familiar with one another. This a priori information mitigates the risk of adverse selection and improves the group quality. This peer selection leverages social capital and networked relationships to mitigate information asymmetry and transaction costs, [44]. Community lending and saving participation signals good repayment behaviour. Group members are selected based on homogeneity of risk profiles, business correlations, and social ties, ensuring low-risk group composition, [45]. Theoretical foundations in the domain of information asymmetry in the context of self-help groups and community lending groups are given below:

Table 2 provides the theoretical frameworks in information asymmetry as challenge to financial inclusion. Despite the success narratives, the self-help groups are fraught with various limitations, such as the risk of default. Such as the risk of overlapping loans, strategic defaults, and use of loans for income smoothing purposes, [46]. Earlier community lending models were devoid of any social contract and joint liability. Self-help groups operate based on joint liability with dynamic incentives for additional bank loans contingent on good repayment behaviour, [47]. They operate through peer monitoring, peer selection, and higher repayment rates mitigate strategic default, [27]. Social capital motivates members to monitor one another and sanction defaulters within homogeneous groups. [48] highlight that the influence of peer monitoring and peer pressure on loan repayment is not uniform and differs with the size of the group, loan size, demographics, including literacy levels, and gender, [24]. Thus, although the theory suggests that peer mechanisms help improve the repayment rates and financial sustainability of the groups, the empirical evidence on the impact of peer mechanisms on repayment behavior in the group is extremely limited. There is a research gap regarding what factors influence the impact of the peer mechanism on loan repayments in a community lending group. Fig 1 depicts how peer selection, peer monitoring and peer sanction impact information asymmetry, in the form of moral hazard and adverse selection.

Fig 1 provides the conceptual framework for reducing information asymmetry

H0: Peer mechanism does not impact the group quality

H1: Peer mechanism has no impact on adverse selection

H2: Peer selection and monitoring do not have an impact on internal loss default

Table 2. Theoretical frameworks in information asymmetry as a challenge to financial inclusion (Source: Author's own work).

| Period | Key theoretical development | Application to SHGs/ Microfinance | Representative works |
|---|---|---|---|
| 1970s (Foundations) | Information asymmetry formalised: adverse selection (hidden type), moral hazard (hidden action). Markets can fail. | Explains why the poor are excluded from formal credit markets; banks ration credit. | Akerlof (1970) *Market for Lemons*; Stiglitz & Weiss (1981) *Credit Rationing* |
| Late 1970s–1980s (Practice Emerges) | Practical innovations in group-based lending: solidarity groups, joint liability. | Grameen Bank (Bangladesh) shows high repayment without collateral, the motivation theory. | Yunus (Grameen Bank, 1976–83) |
| 1990s (Formal Microfinance Theory) | Joint liability and "social collateral" models. Peer monitoring, peer selection, and risk sharing reduce information asymmetry. | SHGs use local knowledge to screen risky borrowers; group enforcement replaces formal collateral. | Besley & Coate (1995). Ghatak (1999) |
| 2000s (Expansion & Optimism) | The "microfinance promise": scaling up SHGs/MFIs as tools for poverty alleviation. | Emphasis on outreach, high repayment, and assumed poverty reduction. | Morduch (1999) *Microfinance Promise* |
| 2010s (RCT Evidence & Critique) | Rigorous impact evaluations question transformative poverty effects. Focus on heterogeneity and context. | SHGs improve access to credit and business activity; poverty/consumption effects are mixed. | Banerjee, Duflo, Glennerster & Kinnan (2015); 6-country RCTs |
| 2025 (Empirical study) | Impact and evaluation of microfinance on poverty alleviation | Positive impact of microfinance on the livelihood of the poor sections of society | Dhungana et al. (2025), |

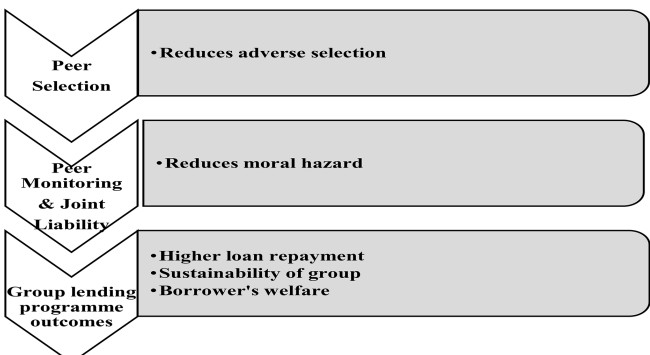

**Fig 1. Conceptual framework for reducing information asymmetry (Source: Author's own work).**

H3: Peer monitoring & selection does not have an impact on external loan default

H4: Peer monitoring and selection do not impact misuse of funds

Despite various case studies on how peer mechanisms have ensured higher repayment rates, certain case groups have disintegrated due to excessive loan burdens. One notable example of community lending failure is the overindebtedness and loan default of microborrowers with Grameen Bank. Similarly, many self-help groups, linked with microfinance institutions, faced repayment defaults in Andhra Pradesh. In the case of APMAS (Andhra Pradesh Mahila Abhivruddhi Society), heavy reliance on joint liability created peer pressure, leading to political backlash, political tensions, and suicides. While ROSCAs thrive on trust, the primary reason for their failure was the informal institutional framework and the lack of legal recourse. Similarly, wealthier members in the village dominated decision-making, leading to the failure of community lending initiatives. Thus, internal factors influence how peer pressure affects repayment within a group. Any heterogeneity among the group members regarding financial literacy, education, social status, and peer pressure might not work equally, and loan default might rise, [49]. Additionally, external factors, such as leadership, may also influence how peer pressure affects loan repayment and default. Another exogenous factor impacting loan repayment and default is digital technology, [50]. Thus, no empirical evidence exists to suggest that peer pressure has an equal impact in different group conditions. Moreover, this research study explores the impact of various factors that account for why peer pressure does not work equally, leading to differences in loan repayment.

## 3. Research methodology

This study employs a quantitative research design to investigate the determinants of group quality (riskiness) as perceived by group members. The objective of the study is to assess the impact of peer mechanisms on internal loan defaults, external loan defaults, and misuse of funds. The primary data was collected from 400 female members of self-help groups in West Bengal through random sampling. For the purpose of data collection, a standardised 29-question questionnaire was validated through a pilot test and follow-up interviews to minimise measurement error. Informed consent was obtained from all participants.

Group quality (riskiness in loan repayment) was measured on a 5-point likert scale. The main independent variable is the peer mechanism, and other factors impacting heterogeneity include (financial literacy, education, social status) and leadership. The empirical strategy involved classifying the groups on the basis of risk propensity using K-means clustering, and the results were corroborated using one-way ANOVA. The core impact analysis was conducted using ordered logistic regression. The study aims to empirically address the research questions regarding the impact of peer

mechanisms on (1) Adverse selection, (2) Moral hazard internal loan default, (3) External loan default, and (4) Misuse of funds. The hypothesis and the empirical strategy for the study are given below:

$$\text{Log} \left( \frac{P}{1-P} \right) = \beta_0 + \beta_1 X_1 + \beta_2 X_2 + \ldots\ldots\ldots \beta_K X_K$$

The study has developed four models that include (1) Adverse selection, (2) Internal delinquency, (3) Misuse of funds, and (4) External delinquency.

> *Hypothesis 1: Peer selection through a priori information about the credit behaviour of other members reduces adverse selection in the group*
>
> *Hypothesis 2: Repayment of internal loans is impacted by peer mechanisms (Peer selection, peer sanction, peer pressure, peer monitoring through meetings, leadership)*
>
> *Hypothesis 3: Repayment of external loans is impacted by peer mechanisms (Peer selection, peer sanction, peer pressure, peer monitoring through meetings, leadership)*
>
> *Hypothesis 4: Misuse of funds is impacted by peer mechanisms (Peer selection, peer sanction, peer pressure, peer monitoring through meetings, leadership)*

### Ethical consideration

The data for the study was collected with the consent of the participants by the qualified researcher/leader in the self-help promoting institution. Due consideration was given to the sensitivity of the participants and the consent was taken from the participants verbally. There were no minors involved in the study and the consent was winessed by the an NGO representative, who was present throughout the data collection. The data were collected verbally and witnessed and documented by the research assistant with SHPI (Self-Help Promoting Institution). And the phone numbers and contact details of the participants were also give. For their time all participants were given a nominal amount to compensate for their loss of work during work hours. Moreover, none of the participants were contacted during their work hours or on official holidays.

## 4. Results and discussions

### Results of ANOVA

The results of the ANOVA analysis highlight important differences among members in terms of risk propensity. In terms of income growth after joining the group, after self-help group membership, this is not uniform across all groups. The F-statistic for the difference among groups is $F = 4.248$, with a significant p-value of 0.015. This implies that while some members report substantial improvement, others perceive little change in income growth. In terms of business stability, there is no significant variation across the clusters ($F = 0.006$, $p = 0.994$). Hence, this question has been dropped. Regarding preference, there is a substantial difference between the clusters. Members' preference for unsecured loans shows a strong difference across the clusters ($F = 3.759$, $p = 0.000$), indicating that while some members are comfortable with taking unsecured loans, others refrain from taking unsecured loans. Similarly, tolerance for paying higher transaction costs, the member's choices differ across the clusters with $F = 17.76$, $p = 0.000$. There are also significant differences among the clusters in terms of preference for a higher interest rate ($F = 316.25$, $p = 0.000$). Finally, preferences for long-term loans show significant variation across clusters. Overall, the findings suggest that there is a significant difference across the clusters in terms of their propensity to assume risk.

Table 3 provides the results of ANOVA. It indicates that a statistically significant difference exists in how members perceive their income after joining the group. Among clusters, there is a very strong and significant difference in preference for unsecured lending. With respect to their willingness to pay higher transaction cost and their tolerance for a high rate of interest, significant differences exist across the clusters. In addition, members demonstrate extremely strong and significant differences in their preference for loan tenure, suggesting diverse borrowing needs.

Table 4 provides the results of cluster analysis. The ANOVA results show a statistically significant difference across the clusters. Table 4 indicates the results of cluster analysis. The sample was divided into 3 clusters (Cluster 1, Moderate Risk = 101 members), Cluster 2 (Low risk = 211) and Cluster 3 (High risk = 88).

Table 5 provides the result of impact of peer mechanism on group quality. As per Table 5, financially literate members have a 12% predicted probability of being all risky. Financial literacy reduces the probability of being in the moderate

**Table 3. Results of ANOVA analysis (Source: Author's own calculation).**

| | Cluster Mean Square | DF | Error Mean Square | DF | F Statistics | Sig |
|---|---|---|---|---|---|---|
| **How do members rate the growth of income after SHG Membership?** | 1.718 | 2 | 0.404 | 397 | 4.248 | 0.015 |
| 1 = Decreased significantly | | | | | | |
| 2 = Decreased somewhat | | | | | | |
| 3 = Same | | | | | | |
| 4 = Increased somewhat | | | | | | |
| 5 = Increased significantly | | | | | | |
| **What is the borrower's assessment of the stability of his business project?** | 0.002 | 2 | 0.292 | 397 | 0.006 | 0.994 |
| **Do you prefer unsecured loans?** | 32.648 | 2 | 0.868 | 397 | 37.59 | 0.000 |
| 1 = Strongly Disagree | | | | | | |
| 2 = Disagree | | | | | | |
| 3 = Same | | | | | | |
| 4 = Agree | | | | | | |
| 5 = Strongly Agree | | | | | | |
| **Do not mind paying high transaction costs?** | 10.20 | 2 | 0.574 | 397 | 17.76 | 0.000 |
| 1 = Strongly Disagree | | | | | | |
| 2 = Disagree | | | | | | |
| 3 = Same | | | | | | |
| 4 = Agree | | | | | | |
| 5 = Strongly Agree | | | | | | |
| **Do not mind paying high-interest rates?** | 111.51 | 2 | 0.353 | 397 | 316.25 | 0.000 |
| 1 = Strongly Disagree | | | | | | |
| 2 = Disagree | | | | | | |
| 3 = Same | | | | | | |
| 4 = Agree | | | | | | |
| 5 = Strongly Agree | | | | | | |
| **Do members prefer long-term loans?** | 176.67 | 2 | 0.586 | 397 | 301.57 | 0.000 |
| 1 = Strongly Disagree | | | | | | |
| 2 = Disagree | | | | | | |
| 3 = Same | | | | | | |
| 4 = Agree | | | | | | |
| 5 = Strongly Agree | | | | | | |

**Table 4. Results of cluster analysis (Source: Author's own calculation).**

| Types of cluster | Number of clusters |
|---|---|
| Cluster 1 (Moderate Risk) | 101 |
| Cluster 2 (Low risk) | 211 |
| Cluster 3 (High risk) | 88 |

**Table 5. Impact of Peer mechanism, Peer selection & Peer sanction on Group Quality (Source: Author's own work).**

| | Coefficients | Probability (Group=1=All Risky) | Probability (Group=Most Risky) | Probability (Moderate Risk) | Probability (Safe) |
|---|---|---|---|---|---|
| Financial literacy | −0.56***(0.296) | 0.12***(0.067) | 0.008(0.006) | −0.071**(0.038) | −0.06***(0.03) |
| Peer sanction | −0.41***(0.188) | 0.08***(0.038) | 0.014(0.008) | −0.051***(0.02) | −0.05***(0.023) |
| Frequency of meetings | 0.48***(0.202) | −0.10***(0.045) | −0.008(0.005) | 0.062***(0.026) | 0.054***(0.020) |
| Access to technology | 0.60***(0.235) | −0.11***(0.042) | −0.029**(0.016) | 0.069***(0.025) | 0.079***(0.035) |
| Leadership | −0.14(0.266) | 0.03(0.056) | 0.004(0.008) | −0.018(0.033) | −0.016(0.031) |
| Moderate risk | 0.32(0.271) | −0.067(0.053) | −0.012(0.013) | 0.039(0.037) | 0.040(0.0355) |
| Low risk | 0.40***(0.242) | −0.086***(0.05) | −0.011(0.007) | 0.051**(0.031) | 0.046***(0.028) |
| Intercept 1 | −0.54(0.344) | | | | |
| Intercept 2 | 0.62(0.346) | | | | |
| Intercept 3 | 2.12(3.64) | | | | |

risk group by 7.1% and the safe group by 6%. Overall, literate individuals are more likely to take more risks. Similarly, strong peer sanctions reduce the probability of being in safe clusters. Peer sanctions have an adverse effect, reducing the likelihood of being in a safe group by 5% and being in a moderate-risk group by 5.1%. Thus, negative reinforcement increases moral hazard and increases the likelihood of being in a risky group. Similarly, frequency of meetings as per peer monitoring reduces the probability of being in all risky groups by 10% and slightly riskier groups by 8%, while increasing the likelihood of being in the moderate risk group by 6.2% and the safe group by 5.4%. Thus, peer monitoring reduces the riskiness of the group. Members with technology access are less likely to be in the risky group by 11% and decrease the likelihood of being in the riskiest group by 2.9%. With access to technology, the likelihood of being in a moderate group is 6.9%, and the likelihood of being in a safe group is 7.9%. Technology strongly encourages safer behaviour. Leadership shows no significant impact on the likelihood of being risky. Being a moderately risky customer increases the likelihood of being moderate risk (3.9%) or safe (4%). Low-risk members are less likely to be in any high-risk group, slightly riskier by 1.1%, and more likely to be in the moderate group by 5.1%. Additionally, they are slightly riskier by 1.1%. They are also more likely to be in the moderate group by 5.1% and are considered safe (4.6%).

The low-risk members (proxy for peer selection) are significantly less likely to be in the high-risk member group and significantly more likely to be in the safe member group. This confirms strong peer selection, which leads to a reduction in adverse selection among group members. Hence, hypothesis 1 (H1) that Peer selection through a priori information about the credit behaviour of other members reduces adverse selection in the group is accepted.

Table 6a provides the results of logistic regression for Model 1, Model 2 and Model 3. According to Table 6a, in Model I, leadership reduces the likelihood of internal loan default, as indicated by a negative β coefficient of −0.462, which is highly significant. Access to technology is negative and significant, with a β coefficient of −0.742, at a high significance level, suggesting that technology reduces the risk of internal loan default. Low-risk members are less likely to default internally compared to high-risk members, with a β coefficient of −0.665 and a high significance level. Other variables, peer pressure, peer sanction, and financial literacy, are not statistically significant. Internal loan default rates are reduced by effective leadership, access to technology, and a low-risk member composition.

**Table 6. (a) Results of logistic regression for Model 1, Model 2, Model 3 (Source: Author's own calculation). (b): Marginal effects of peer mechanism on internal default, external default, and misuse of funds. (Source: Author's own work).**

| | Internal default | External default | Misuse of funds |
|---|---|---|---|
| **(a) Results of logistic regression for Model 1, Model 2, Model 3** | | | |
| Peer pressure | −0.138(0.298) | 0.085(0.29) | −0.525***(0.302) |
| Peer sanction | 0.295(0.294) | 0.278(0.286) | 0.713***(0.301) |
| Willingness to help others | | −0.405***(0.223) | −0.141(0.240) |
| Frequency of meetings | −0.240(0.241) | −0.480***(0.231) | 0.177(0.155) |
| Progressive lending | | | |
| Leadership | −0.462***(0.227) | −0.214(0.220) | |
| Income | | 0.221(0.183) | |
| Financial Literacy | −0.088(0.277) | −0.055(0.233) | −0.869***(0.273) |
| Education | −0.064(0.302) | | −0.854***(0.304) |
| Access to technology | −0.742***(0.281) | | 0.756(0.284) |
| Business Correlation | −0.585(0.958) | | 0.362(0.124) |
| Risk Type | | | |
| Moderate Risk | 0.065(0.341) | −0.932***(0.324) | −0.907***(0.349) |
| Low Risk | −0.665***(0.293) | −0.060(0.275) | −0.961***(0.378) |
| Constant | 1.46***(0.379) | −0.164(0.502) | 1.509***(0.395) |
| Negelkerke R Square | 0.110 | 0.078 | 0.173 |
| **b. Marginal effects of peer mechanism on internal default, external default, and misuse of funds** | | | |
| | Internal loan default | External loan default | Misuse of funds |
| Peer pressure | −0.031(0.068) | 0.020(0.069) | −0.127***(0.073) |
| Peer sanction | 0.067(0.067) | 0.066(0.068) | 0.172***(0.073) |
| Willingness to help others | | −0.096**(0.053) | |
| Frequency of meetings | −0.055(0.055) | −0.114***(0.055) | −0.034(0.058) |
| Progressive lending | | | 0.043(0.037) |
| Leadership | −0.105***(0.051) | −0.051(0.0522) | |
| Income | | 0.052(0.0438) | |
| Financial Literacy | −0.020(0.063) | −0.013(0.055) | −0.210***(0.066) |
| Education | −0.014(0.069) | | −0.207***(0.073) |
| Access to technology | −0.158***(0.064) | | 0.018(0.068) |
| Business Correlation | −0.133(0.219) | | 0.087(0.272) |
| Risk Type | | | |
| Moderate Risk | 0.015(0.078) | −0.222***(0.077) | −0.219***(0.084) |
| Low Risk | −0.152***(0.066) | −0.014(0.065) | −0.232***(0.091) |
| Constant | | | |

Hence, the H2 is partially supported, indicating that peer selection reduces internal loan default repayment. Low-risk members in a group reduce the probability of internal loan default. Peer sanction and peer pressure do not have an impact on internal loan default and repayment.

In Model II, the analysis of the model reveals that the willingness to help others is negative and significant, with a coefficient of β equal to −0.48 and a high level of significance, suggesting that regular meetings strengthen repayment discipline towards external lenders. Moderate-risk members are less likely to default externally compared to high-risk members, with a β coefficient of −0.932 and a high level of statistical significance. Other variables such as literacy, education,

leadership, and peer effects are not significant. External loan default or default on bank loans is significantly reduced by cooperative behaviour, frequent meetings and having few high-risk members.

From the analysis of the statistical results, strong leadership significantly reduces internal loan default, and peer selection also reduces the risk of internal default significantly. This leads to the rejection of the hypothesis that peer selection significantly reduces the risk of selecting low-risk members.

In model III, peer pressure is negative and significant, with a β coefficient of −0.525 and a high level of significance, indicating that peer pressure reduces the likelihood of misuse of funds. Peer sanction has a β coefficient of 0.713 and a high level of significance, indicating that peer sanction or negative reinforcement increases the likelihood of misuse of funds. Financial literacy, with a β coefficient of −0.869 and a high significance level, indicates that literate members are less likely to misuse funds. The education coefficient is negative and significant, with a β of −0.854 and a high significance level, indicating that higher education reduces the likelihood of fund misuse. The business correlation coefficient β = 0.362 is also positive and significant, indicating that a positive business correlation increases the likelihood of default. Both moderate- and low-risk members are less likely to misuse funds, with β coefficients of −0.907 and −0.961, respectively, and a high level of significance. Misuse of funds is significantly reduced by financial literacy, education, peer pressure, and moderate and low risk. While peer sanctions and business correlation increase the probability of misuse of funds.

Based on the analysis, hypothesis 3 is accepted that peer pressure reduces the misuse of funds. And low-risk and moderate-risk members reduce the instances of misuse of funds.

Table 6b provides the marginal effects of peer mechanism. Internal default risk is statistically reduced by leadership quality, access to technology, and the presence of low-risk members. Leadership reduces the probability of internal loan default by 15.8%. And low-risk members are 15.2% less likely to default internally. External loan default is reduced by poor monitoring and members in low-risk categories. Willingness to help others reduces the probability of default by 9.6% and frequency of meetings reduces the probability of default by 11.4%. Moderate-risk members are 22.2% less likely to default externally compared to high-risk members. Table 4(b) provides the results of logistic regression.

To classify borrowers into various risk types, cluster analysis was employed following one-way ANOVA. There was a significant difference among the risk types, and cluster scores were generated as a proxy of risk class. For generating clusters based on riskiness, the members were classified on the basis of [1] income growth, [2] debt terms. For forming the clusters, the questions used are: [1] How do members rank their income growth after joining the group? And how do they rank the groups based on debt growth and preference of loan covenants, which include [1] maturity of loans, [2] cost of loan, and [3] transaction costs. These findings are further corroborated through the use of One-way ANOVA, and the results show a significant difference among groups. These questions were asked because small entrepreneurs are difficult for formal financial institutions, such as banks, to assess; they often lack information about them. The basic premise is that low-quality borrowers are insensitive to higher borrowing and transaction costs.

## Discussion and analysis

### Hypothesis I: Peer selection through a priori information about the credit behaviour of other members reduces adverse selection in the group

The study substantiates that peer selection mitigates the adverse selection, and this is consistent with earlier findings in the study, [51]. Thus, having prior information about the riskiness of the members enables peer selection and risk matching, thereby reducing the risk of adverse selection. However, the literature mentions the problem of moral hazard or the change in financial behaviour of the members after joining the group. With a priori information, the literature emphasises the problem of strategic default or collusion among the members of the group, [52]. In our analysis, Model II discusses the impact of peer mechanisms on internal and external loan repayment defaults.

Similarly, peer monitoring through frequent meetings increases the likelihood that the group is safe. This is consistent with the findings of the earlier studies, [53]. [54] highlights that when group meetings were reduced from

fortnightly to monthly, repayment delays and defaults increased in the group. Besides that, reducing the number of meetings eroded the group's trust, leading to repayment defaults. Thus, meetings definitely help to improve accountability and trust among the group members. But there is a potential downside: a very high frequency of meetings can lead to group fatigue and impose psychological cost on members by reducing participation quality and imbalance in the family and personal lives of the members in the group, [55]. A more extensive study can be undertaken in the future to account for the influence of a higher number of group meetings and geographical context on loan repayment behaviour and group quality.

An interesting finding is that peer sanctions negatively impact group quality, contrasting with earlier studies' findings, [56]. According to the literature, peer sanctions are implemented to ensure compliance and reduce free-riding behavior among group members. [53] In his study, he has emphasised that peer sanctions are designed to reduce free riding in a group. However, the findings of our analysis indicate that peer sanctions lead to a higher propensity to take risks. Sanctions can erode trust, cohesion, cooperation, and solidarity within a microfinance self-help group. Due to the threat of sanctions, the group members might decline to cooperate and support each other in case of a contingency, [57]. Thus, according to our study, peer sanctions may lead to the erosion of trust and social capital within the group.

External factors such as financial literacy also play a significant role in shaping group quality, aligning the results with the theory of peer monitoring, which emphasises the role of the peer mechanism and reduces adverse selection, [58]. Surprisingly, financial literacy increases the likelihood of risk-taking. This could be due to procyclicity and overconfidence bias, which means that the literate members might take calculated risks, thinking that they can manage risk better, [59]. Alternatively, members with more information and literacy might expose themselves to more borrowing options, making them more vulnerable to credit risk. Due to higher procyclicity and inclination to take risks, the borrowers with higher financial literacy might face more credit risk

**Hypothesis II: Repayment of internal loans is impacted by peer mechanisms (Peer selection, peer sanction, peer pressure, peer monitoring through meetings, leadership)**

Additionally, earlier studies [60] have highlighted that peer sanctions reduce internal loan defaults. Our study, in contrast, argues that peer sanctions are positively associated with internal loan default. A plausible explanation is that cohesive groups and groups with strong leadership are less reliant on negative sanctions to facilitate repayment of loans, [61]. Thus, according to the analysis of the data, strong leadership enhances group cohesion and reduces the need for punitive sanctions. Good leadership in a group fosters a culture of repayment and discipline. Furthermore, although financial literacy negatively impacts internal loan default, the results are not statistically significant. This suggests that although financial literacy may provide a better understanding of financial obligations, it does not necessarily prevent opportunistic behavior. However, the use of technology reduces the misuse of funds by bringing about better disclosure and transparency in the utilisation of funds, [62] The group linkage programme is also supported by various Government and non-government microfinance Institutions. An earlier study highlights the importance of a pluralistic microfinance ecosystem in achieving empowerment goals [27]. This will help to improve the performance of the groups [23]. However, this study is limited to the linkage between self-help groups and banks, and further research can be conducted in this area.

**Hypothesis III: Peer monitoring, sanctions and pressure reduce the moral hazard in a group**

The data analysis reveals that peer monitoring decreases the likelihood of external loan default within a group. However, peer monitoring undertaken through regular meetings and informal interactions promotes greater accountability and financial discipline among group members. Regular meetings and evaluation of repayment schedules reduce the likelihood of default among the members of the group, [60]. Thus, an optimum number of meetings and repayment schedules reduces

the likelihood of external loan default. This is in line with the findings of an earlier study [63] However, an excessive number of meetings results in personal fatigue and psychological costs, which can result in strategic default, [54]. Additionally, the willingness of members to help one another reduces the risk of external default among group members. Willingness to help others leads to higher solidarity, and better cohesion leads to trust and reciprocity among the members of the group, [64]. Indeed, social capital serves as the basis for lending to group members. Thus, social capital improves access to formal finance while reducing external loan default. Several studies emphasize the significance of microfinance and group lending in enhancing the living standards of households. In the context of reduced external loan default, more research can be undertaken to explore the impact of microfinance on household living standards [63].

**Hypothesis IV: Peer pressure and peer sanctions reduce the misuse of funds among the members of the group**

The study's findings show that peer pressure, which utilizes social influence and informal norms, reduces the misuse of funds among group members. This is supported by the findings of an earlier study, [65]. Further, [66] in his analysis has highlighted that the increasing misuse of funds in the case of a large-scale default episode in Andhra Pradesh, India. The study's findings indicate that social pressure and peer pressure do not stabilize repayment behavior during the crisis; rather, they exacerbate the financial plight of the group, leading to widespread defaults. Thus, peer pressure can have negative effects in the event of a crisis. While peer pressure and social influence reduce the misuse of funds in the group, as per [67]. Thus, although our study indicates that social influence and peer pressure reduce the misuse of funds, there is still scope for further study regarding the factors that influence the strength and direction of the relationship between peer pressure and misuse of funds. The size of the group and regional variations might act as moderators in influencing the strength of the relationship between the peer mechanism and misuse of funds. Additionally, an earlier study [24] has highlighted the role of loan size on group productivity, and further research can be conducted in this area.

## 5. Conclusion

According to the study, peer mechanisms are crucial in determining group quality and sustainability. As per the study's findings, peer selection effectively reduces adverse selection, which is consistent with existing theoretical findings; however, the risk of collusion persists. Peer monitoring through regular meetings enhances accountability and fosters repayment discipline within the group. However, excessive frequency of meetings can lead to psychological costs. Contrary to earlier findings, peer sanctions erode trust and cohesion while increasing default risk. Thus, there is a notable finding that several self-help groups employ peer sanctions, which raises serious ethical considerations and could be an area of future research. Rather than peer sanctions, leadership and social capital, in the form of business connections, improve group quality and repayment rates. Hence, more emphasis should be on exercising control through better leadership, whereas the use of informal and peer pressure helps to reduce the misuse of funds, but their impact and effect might vary across contexts. Overall, peer mechanisms such as selection, monitoring and peer pressure can reduce adverse selection and moral hazard; their effectiveness is contextual. The use of technology and effective leadership can reduce default risks and foster trust among group members. Another very interesting finding is that, contrary to expectations that financial literacy reduces risk, financially literate members have a higher predicted probability of being in the risky group. This suggests that the financially literate members are more prone to overconfidence bias and procyclicity. Further research can be undertaken to investigate the impact of various contextual moderators, such as geographical or regional variation, group size, to design more adaptive policy frameworks leveraging peer mechanisms.

## 6. Limitations of study

One of the limitations of the study is that, in its current scope, it could not undertake research on the impact of group heterogeneity in terms of group size, regional variation on loan repayment and the sustainability of the group. It also does not take into account the impact of a higher frequency of meetings on the loan repayment behaviour and group quality.

## 7. Future scope for research

Although this study attempts to measure the influence of various socio-demographics, such as gender, education, and financial literacy, on repayment behavior and group quality. In the future, more research could be undertaken on the impact of regional variation, group size, and group heterogeneity on loan repayments and group quality.. Additional research can be conducted on the impact of a higher frequency of meetings on loan repayment behavior and group quality.

## Supporting information

**S1 File. External loan default Q.**
(CSV)

**S2 File. Internal loan default.**
(CSV)

**S3 File. Misuse of funds.**
(CSV)

**S4 File. Model 1 Dataset.**
(CSV)

## Author contributions

**Conceptualization:** Nishi Malhotra.

**Data curation:** Nishi Malhotra.

**Formal analysis:** Nishi Malhotra.

**Funding acquisition:** Nishi Malhotra.

**Investigation:** Nishi Malhotra.

**Methodology:** Nishi Malhotra.

**Project administration:** Nishi Malhotra.

**Resources:** Nishi Malhotra.

**Software:** Nishi Malhotra.

**Supervision:** Nishi Malhotra.

**Validation:** Nishi Malhotra.

**Visualization:** Nishi Malhotra.

**Writing – original draft:** Nishi Malhotra.

**Writing – review & editing:** Nishi Malhotra.

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
