## [Decision Letter · Decision Letter 0]

17 Sep 2025

PONE-D-25-40003How peer mechanism impacts loan repayment in a Self-help group? An empirical studyPLOS ONE?

Dear Dr. Malhotra,

Thank you for submitting your manuscript to PLOS ONE. After careful consideration, we feel that it has merit but does not fully meet PLOS ONE’s publication criteria as it currently stands. Therefore, we invite you to submit a revised version of the manuscript that addresses the points raised during the review process.

The manuscript has been evaluated by two reviewers, and their comments are available below.

 

We look forward to receiving your revised manuscript.

Kind regards,

Jenna Scaramanga

Staff Editor

PLOS ONE

**Journal Requirements** :

Reviewers' comments:

Reviewer's Responses to Questions

**Comments to the Author**

1. Is the manuscript technically sound, and do the data support the conclusions?

Reviewer #1: Partly

Reviewer #2: No

2. Has the statistical analysis been performed appropriately and rigorously?

Reviewer #1: Yes

Reviewer #2: No

3. Have the authors made all data underlying the findings in their manuscript fully available?

Reviewer #1: Yes

Reviewer #2: Yes

4. Is the manuscript presented in an intelligible fashion and written in standard English?

Reviewer #1: Yes

Reviewer #2: No

Reviewer #1: • Topic: Good, interesting and researchable. But the study location is missing. Better to add at the last of the topic – in India

• Abstract –Lengthy, concise it.

• Introduction: Poor. The contextual issues are missing. Use of old citations (not after 2017). Need to add latest citations. Remove problem statements and objectives of the research under the sub-headings (It is not a thesis, a research paper!). Keep it under the introduction part in a concise form.

• Literature Review: Poor. The theory, model and policy related to the research area is missing. Not shown the research gap and a conceptual framework. The review of paper is too old, not found citations after 2015. The way of citations is not proper. Significant improvement is required. Consult the papers and add the latest citations:

Dhungana, B. R., Chapagain, R., Pokhrel, O. P., Sharma, L. K., & Gurung, J. B. (2025). Assessing the performance of microfinance institutions in Nepal. Pacific Business Review International, 17(11).

Dhungana, B. R., Chapagain, R., & Ashta, A. (2023). Alternative strategies of for-profit, not-for-profit and state-owned Nepalese microfinance institutions for poverty alleviation and women empowerment. Cogent Economics & Finance, 11(2), 2233778.

Chapagain, R. K., & Dhungana, B. R. (2020). Does microfinance affect the living standard of the household? Evidence from Nepal. Finance India, 34(2).

Dhungana, B. R. (2016). Does loan size matter for productive application? Evidence from Nepalese micro-finance institutions. Repositioning:The Journal of Business and Hospitality, 1, 63-72.

• Methodology: Poor. No need sub headings. Concise it and remove irrelevant writings.

• Results and Discussion: Poor. This section is missing. You’re your results and analysis under this section. You need to compare your results with previous researchers that may be for and against with your findings. It is necessary to cite properly.

• Conclusion: Concise it. Highlight what interesting message you found and provide suggestions to the policy makers. Also highlight -why your research is useful for policy implications.

• Others: Follow the format as prescribed by the journal.

Reviewer #2: The introduction lacks a broader context, failing to connect the study to the wider global landscape of microfinance and self-help groups (SHGs). It could also provide a more detailed discussion on how the problem of information asymmetry manifests in various microfinance models.

The Literature Review section primarily relies on foundational theories without exploring more recent or conflicting perspectives. It also lacks a critical review of the limitations of the existing research it cites, such as contexts where peer mechanisms have failed.

Findings and Discussion: The discussion is confirmatory mainly and does not deeply explore counterintuitive or unexpected findings. It lacks a nuanced analysis of potential downsides, such as whether a high frequency of meetings could lead to group fatigue. Comparing the findings to other geographical or cultural contexts is also missing.

Conclusion and Implication: This section lacks a discussion of the policy and practical limitations of the study's findings. It recommends the peer mechanism for financial inclusion but doesn't address the real-world challenges of large-scale implementation, such as training costs or the ethical implications of social sanctions.

Overall, the paper is also poorly written, as it lacks a research method, and the numbering is inconsistent.

**Do you want your identity to be public for this peer review?** For information about this choice, including consent withdrawal, please see our Privacy Policy

Reviewer #1: **Yes:** Bharat Ram Dhungana

Reviewer #2: No

---

## [Author Response · Author response to Decision Letter 1]

14 Oct 2025

Reply to queries

Thanks for your valuable review comments. I am highly obliged and thankful for your consideration and time. I have incorporated the comments provided by Reviewer 1 and Reviewer 2. I am pleased to submit the manuscript for your consideration.

REVIEWER 1

Comment 1: Abstract –Lengthy, concise it.

Reply: I have synthesised the abstract and made it concise. The revision has been made in the manuscript

Section 1 “Women’s empowerment…………………………research.”

Comment 2: Introduction. The contextual issues are missing. Use of old citations (not after 2017). Need to add the latest citations. Remove problem statements and objectives of the research under the subheadings (It is not a thesis, a research paper!). Keep it under the introduction part in a concise form.

“Globally…………………………………………………………………………. hazard”

Reply: I have added the latest citations in the study. And the problem and objectives of the research under subheadings have been removed. At the same time, I have added the contextual issues in the introduction of the study.

Comment 3: Literature: The theory, model and policy related to the research area are missing. The research gap and a conceptual framework are not shown. The review of the paper is too old, and it does not cite citations after 2015. The way of citations is not proper. Significant improvement is required. Consult the papers and add the latest citations:

Dhungana, B. R., Chapagain, R., Pokhrel, O. P., Sharma, L. K., & Gurung, J. B. (2025). Assessing the performance of microfinance institutions in Nepal. Pacific Business Review International, 17(11).

Dhungana, B. R., Chapagain, R., & Ashta, A. (2023). Alternative strategies of for-profit, not-for-profit and state-owned Nepalese microfinance institutions for poverty alleviation and women empowerment. Cogent Economics & Finance, 11(2), 2233778.

Chapagain, R. K., & Dhungana, B. R. (2020). Does microfinance affect the living standard of the household? Evidence from Nepal. Finance India, 34(2).

Dhungana, B. R.(2016). Does loan size matter for productive application? Evidence from Nepalese micro-finance institutions. Repositioning: The Journal of Business and Hospitality, 1, 63-72

Conceptual framework and research gap have been included in the literature review section of the study

Reply: I have added the latest citations, and I have also included the research gap and conceptual framework in the data. I have tried to update the literature review as per the latest developments in the field. The comments are provided in the study along with the changes.

Comment 4: Methodology: No need for subheadings. Concise it and remove irrelevant writings

Reply: Irrelevant writings have been removed from the methodology section.

Comment 5: Result & analysis: This section is missing. You’re your results and analysis under this section. You need to compare your results with previous researchers who may be for or against your findings. It is necessary to cite

Reply: Results and analysis have been improved. The findings from the other study have corroborated the results.

Under the discussion and analysis section for all the hypotheses, the findings from other studies have been included to corroborate the findings and the results

Comment 6: Concise it. Highlight what interesting message you found and provide suggestions to the policy makers. Also, highlight why your research is helpful for policy implications

Reply: In the conclusion section, I have included the policy implications and new findings of the study

Others: Follow the format as prescribed by the journal

Reply: Format prescribed by the journal has been followed

REVIEWER 2:

Comment 1: The introduction lacks a broader context, failing to connect the study to the wider global landscape of microfinance and self-help groups (SHGs). It could also provide a more detailed discussion on how the problem of information asymmetry manifests in various microfinance models.

Reply: I have tried to add the contextual details in the introduction of the study. I have also added the details regarding the microfinance models in the study.

“Globally………………..hazard”

Comment 2: The Literature Review section primarily relies on foundational theories without exploring more recent or conflicting perspectives. It also lacks a critical review of the limitations of the existing research it cites, such as contexts where peer mechanisms have failed.

Reply: I have tried to incorporate the conflicting perspectives in the study, and also included the critical review and limitations of the study in the data. I have also added the details regarding the instances where the peer mechanism has failed in the past

“Despite various case studies on how peer mechanisms have ensured higher repayment, certain case groups disintegrated due to excessive loan burden. One of the notable examples of community lending failure is over indebtedness and loan default of micro borrowers with Grameen Bank. Similarly, many self-help groups, linked with microfinance institutions, faced repayment defaults in Andhra Pradesh. In the case of APMAS (Andhra Pradesh Mahila Abhivruddhi Society), heavy reliance on joint liability created peer pressure, leading to political backlash, political tensions, and suicides. While ROSCAs thrive on thrust, the reason for failure was the informal institutional framework and lack of legal recourse. Similarly, wealthier members in the village dominated decision-making, leading to the failure of community lending initiatives. Thus, internal factors impact how peer pressure impacts repayment in a group. Any heterogeneity among the group members regarding financial literacy, education, social status, and peer pressure might not work equally, and loan default might rise (Verma et al., 2024). Besides, the external factors, like leadership, might also impact how peer pressure influences loan repayment and default. Another exogenous factor impacting loan repayment and default is digital technology. Thus, no empirical evidence exists that peer pressure works equally in different group conditions. Moreover, this research study explores the impact of various factors that account for why peer pressure does not work equally, leading to differences in loan repayment.”

Findings and Discussion: The discussion is mainly confirmatory and does not deeply explore counterintuitive or unexpected findings. It lacks a nuanced analysis of potential downsides, such as whether a high frequency of meetings could lead to group fatigue. Comparing the findings to other geographical or cultural contexts is also missing.

Reply: In the findings and discussion section, I have included the potential downside, such as whether a higher frequency of meetings could lead to group fatigue.

Hypothesis 3: Peer monitoring, sanctions and pressure reduce the moral hazard in a group

The data analysis highlights that peer monitoring reduces the likelihood of external loan default in a group. However, peer monitoring that is undertaken through regular meetings and informal interaction promotes more accountability and financial discipline among the members of the group. Regular meetings and evaluation of repayment schedules reduce the likelihood of default among the members of the group (Tri, 2024). Thus, an optimum number of meetings and repayment schedules reduces the likelihood of external loan default. However, an excessive number of meetings results in personal fatigue and psychological costs, which can result in strategic default (Pellegrina et al, 2021). In addition to this, the willingness of members to help other members of the group reduces the risk of external default among the members of the group. Willingness to help others leads to higher solidarity, and better cohesion leads to trust and reciprocity among the members of the group (Feigenberg, Field, & Pande, 2019). Indeed, social capital serves as the basis of lending to the members of the group. Thus, social capital improves access to formal finance while reducing external loan default.

Conclusion and Implication: This section lacks a discussion of the policy and practical limitations of the study's findings. It recommends the peer mechanism for financial inclusion but doesn't address the real-world challenges of large-scale implementation, such as training costs or the ethical implications of social sanctions.

Reply: In the conclusion, I have tried to add the details regarding the increasing training costs and the ethical implications of the social sanctions

As per the study, peer mechanisms are critical in determining group quality and sustainability. According to the findings of the study, peer selection effectively reduces adverse selection, which is consistent with existing theoretical findings, though the risk of collusion persists. Peer monitoring through regular meetings enhances accountability and brings about repayment discipline in the group. However, excessive frequency of meetings can lead to psychological costs. Contrary to earlier findings, peer sanctions erode trust and cohesion while increasing default risk. Thus, there is a stark finding that several self-help groups use peer sanctions, which raises serious ethical considerations and can be an area of research for the future. Rather than the findings that the leadership and social capital in the form of business correlation improve group quality and repayments, more emphasis should be on exercising control through better leadership, whereas the use of informal and peer pressure helps to reduce the misuse of funds, but their impact and effect might vary across contexts. Overall, peer mechanisms such as selection, monitoring and peer pressure can reduce adverse selection and moral hazard; their effectiveness is contextual. Use of technology and appropriate leadership can reduce default risks and sustain trust among the group members. Further research can be undertaken to investigate impact of various contextual moderators, such as geographical or regional variation, group size, to design more adaptive policy frameworks leveraging peer mechanisms

---

## [Decision Letter · Decision Letter 1]

15 Dec 2025

PONE-D-25-40003R1How peer mechanism impacts loan repayment in a self help group ? An empirical study in IndiaPLOS One?

Dear Dr. Malhotra,

Thank you for submitting your manuscript to PLOS ONE. After careful consideration, we feel that it has merit but does not fully meet PLOS ONE’s publication criteria as it currently stands. Therefore, we invite you to submit a revised version of the manuscript that addresses the points raised during the review process.

Based on the reviewer’s final comments, I am pleased to inform you that your manuscript is accepted pending minor revisions.

Please ensure the following before final acceptance:

Add a few recent and relevant citations in the discussion section, as suggested by the reviewer.Ensure that the manuscript fully adheres to the journal’s formatting guidelines.

Once these minor adjustments are made, we will proceed with the final steps of publication.

We look forward to receiving your revised manuscript.

Kind regards,

Zakaria Boulanouar, PhD

Academic Editor

PLOS One

Journal Requirements:

Additional Editor Comments:

Based on the reviewer’s final comments, I am pleased to inform you that your manuscript is accepted pending minor revisions.

Please ensure the following before final acceptance:

1- Add a few recent and relevant citations in the discussion section, as suggested by the reviewer.

2- Ensure that the manuscript fully adheres to the journal’s formatting guidelines.

Once these minor adjustments are made, we will proceed with the final steps of publication.

Reviewers' comments:

Reviewer's Responses to Questions

**Comments to the Author**

Reviewer #1: All comments have been addressed

Reviewer #2: All comments have been addressed

2. Is the manuscript technically sound, and do the data support the conclusions?

Reviewer #1: Yes

Reviewer #2: Yes

3. Has the statistical analysis been performed appropriately and rigorously?

Reviewer #1: Yes

Reviewer #2: Yes

4. Have the authors made all data underlying the findings in their manuscript fully available?

Reviewer #1: Yes

Reviewer #2: Yes

5. Is the manuscript presented in an intelligible fashion and written in standard English?

Reviewer #1: No

Reviewer #2: Yes

Reviewer #1: (No Response)

Reviewer #2: As this is the second review, the author has addressed few comments and revised it accordingly. Hence, the article could be considered for publication

**Do you want your identity to be public for this peer review?** For information about this choice, including consent withdrawal, please see our Privacy Policy

Reviewer #1: **Yes:** Bharat Ram Dhungana

Reviewer #2: No

---

## [Author Response · Author response to Decision Letter 2]

22 Dec 2025

RESPONSE TO REVIEWERS

1. Addition of Recent and Relevant Citations: As suggested by you, the comments have been included in the manuscript (Discussion section) and given as references 23, 27, 64, and 24. Citations suggested by the reviewer have been added

2. Adherence to Journal Formatting Guidelines:

The manuscript has been thoroughly revised, and all necessary changes have been implemented.

1) Reference style has been changed to Vancouver

2) Double spacing and line numbers have been added

I believe that these minor revisions have improved the overall clarity and scholarly contribution of the manuscript. We hope that the revised version meets the journal’s expectations and is now suitable for final acceptance.

Regards

Dr Nishi Malhotra

---

## [Editor Report · Decision Letter 2]

11 Jan 2026

How peer mechanism impacts loan repayment in a self help group ? An empirical study in India

PONE-D-25-40003R2

Dear Dr. Malhotra,

We’re pleased to inform you that your manuscript has been judged scientifically suitable for publication and will be formally accepted for publication once it meets all outstanding technical requirements.

Kind regards,

Zakaria Boulanouar, PhD

Academic Editor

PLOS One

Additional Editor Comments (optional):

Dear Dr. Malhotra,

We are pleased to inform you that your manuscript has been accepted for publication in the journal.

The editorial office will contact you in due course regarding the next steps in the production process.

Thank you for your interest in and contribution to the journal.

Regards
---

## [Editor Report · Acceptance letter]

PONE-D-25-40003R2

PLOS One

Dear Dr. Malhotra,

I'm pleased to inform you that your manuscript has been deemed suitable for publication in PLOS One. Congratulations! Your manuscript is now being handed over to our production team.

Kind regards,

on behalf of

Dr. Zakaria Boulanouar

Academic Editor

PLOS One